# Contextual Variables with an Impact on the Educational Inclusion of Students with Rare Diseases

**DOI:** 10.3390/ijerph192114103

**Published:** 2022-10-28

**Authors:** Ramón García-Perales, Ascensión Palomares-Ruiz, Andrea Gracia-Zomeño, Eduardo García-Toledano

**Affiliations:** Faculty of Education of Albacete, University of Castilla-La Mancha, 02071 Albacete, Spain

**Keywords:** rare disease, educational inclusion, teacher, association, ownership, location/setting

## Abstract

The context of a school may play a fundamental role in students’ academic and personal progress. In this study, we focus on two contextual variables, the school type and school location or setting. The study used a questionnaire to assess teachers’ knowledge and thoughts about rare diseases based on these variables, with the participation of 574 school teachers. To broaden the research perspective, another questionnaire was administered to members of 152 rare disease patient advocacy groups to ask about their participation in educational processes and analyse their results according to one of the contextual variables: the setting or location of each association. The results indicated statistically significant differences according to the variables examined, which were larger for the type of school variable. In short, numerous variables that influence the teaching and learning processes need to be considered in educational praxis; in this study, we looked at those of a contextual nature (for example, the geographic characteristics of schools and associations), and this is essential for increasingly heterogeneous educational locations that demand multidimensional approaches.

## 1. Introduction

Inclusion benefits everyone involved in education, regardless of the context, existing characteristics, circumstances [1,2], or potential obstacles during schooling [3]. It is fundamental in delivering an education that is tailored to each student, always mindful of the need for collaborative, multidimensional work addressing all the contexts surrounding the student, work that is shared between all the teachers involved [4]. One of the areas that requires specific training is the application of technology to school inclusion [5,6], considering the numerous benefits to student integration [7,8].

It is essential for teaching and learning processes to be applied generally—without any kind of discrimination—focused on individuality, tolerance, and freedom, tailoring education to each student’s characteristics and potential. This will promote the full development of each individual’s personality along with the values, habits, and attitudes that will benefit everyone, as inclusion is a fundamental premise of a contemporary educational system [1,9]. School must be an integrating space that promotes opportunities to live and learn, become a meeting place with diverse experiences and take equity and inclusion as the essential principles of action with all students [3].

Many students have particular educational needs and require specific measures and resources to ensure their inclusion in education. This includes students suffering from rare diseases [10]. ‘Rare diseases’ is a complex term to define conceptually because of the variety of diseases, the low frequency, the severity, some have a high mortality [11], and the availability of treatment; many of these diseases are considered “orphans” due to a lack of pharmacological treatments [12,13,14]. Most appear in young children because they are predominantly genetic [15], which only heightens the importance of addressing them educationally and socially from an early age.

In education, some level of appropriate knowledge about these diseases is essential in teaching [16,17]. Professionals whose work is providing specific support to these students play a key role in guiding and advising other members of the teaching team, families, and social institutions that have an impact on students’ educational progress. These include school guidance counsellors, who play an essential role in counselling, guidance, mediation, and coordination to help all students adjust right from the beginning of their schooling [18]. Guidance counsellors’ work in all of the contexts surrounding students is fundamental in helping them to develop comprehensively, allowing them as balanced a transition between educational stages as possible [19]. Advice and guidance are not solely aimed at students with rare diseases; the essence of this work also covers all associated areas, for example, sometimes these students’ families may exhibit a certain amount of dissatisfaction with the care and support they receive [20].

There are many contextual elements in students’ schooling. In this study, we focus on contextual school characteristics, looking at the type of school and the location/setting. This socio-educational approach is essential to the well-being of these students with rare diseases [21]. These contextual variables could influence teachers in their work, promoting these students’ educational inclusion. Teachers play an essential role in their students’ inclusion [22,23,24,25], a role that is performed in a certain educational and social context.

Consequently, in educational and care work aimed at these children, the social perspective of intervention is essential. This approach is mainly represented by patient support groups (or associations or advocacy groups), whose action is closely linked to school and family contexts. Where these associations are based is important since there are many more in urban than in rural locations, meaning that students who attend schools in smaller towns and villages may be more prone to isolation and a lack of tailored, diversified education. Similarly, emphasizing what was noted previously, technology could play a fundamental role. Integrating technology into education is essential [26,27], for example, online meetings with associations that deal with the same pathology could help address the deficiencies noted above and raise awareness about rare diseases [28].

The variety of rare diseases means that they have been split into various categories, which has had an impact on the creation of patient groups and associations. The diseases have been categorized as follows [29]: diseases of the digestive system; endocrine, nutritional, and metabolic diseases; congenital malformations, deformities, and chromosomal abnormalities; diseases of the musculoskeletal system and connective tissue; nervous system diseases; diseases of the blood and hematopoietic organs; diseases of the ear and mastoid process; tumours (neoplasties); skin and subcutaneous tissue diseases; diseases of the eye and its annexes; injuries, poisoning, and some other consequences of external causes; a disorder of the olfactory and taste system; circulatory system disorders; diseases of the genitourinary system; mental and behavioural disorders; certain infectious and parasitic diseases; rare neurological diseases; certain conditions originating in the perinatal period; diseases of the respiratory system; and unspecified diseases that do not fit into any of the above categories.

This typology determines each association’s primary objectives, and their resources and interventions are tailored accordingly. Because of this variety, and the scarcity of resources and existing funding, volunteers in these associations play a leading role, particularly in schools to bring this support web together with social, cultural, and educational contexts. The United Nations supports volunteering in all spheres of life, considering it a key element in promoting sustainable development, collective action, stronger social bonds and cohesion, as well as inclusion, solidarity, equality, and active citizenship [30]. At the European level, it is worth highlighting the European Solidarity Corps [31], since its priorities include social inclusion, valuing difference and diversity for all, and equal access to opportunities through necessary resources, activities, and support measures. Accepting these goals at the European level is essential to join actions in the implementation of inclusive education in Europe [32]. In Spain, Law 45/2015, of 14 October, on Volunteering [33] includes the within socio-health volunteering actions aimed at social care for society as a whole or directed at specific vulnerable groups—including people with rare diseases [34]—to promote comprehensive development in all areas of people’s lives through support and specialized guidance to improve their conditions and quality of life.

In summary, the objective of this study was to examine the impact of the contextual variables of the school type and setting/location on teachers’ knowledge and thoughts about education for these students. It also examined patient advocacy groups’ opinions of their actions in education—mainly dissemination, publicity, and raising awareness of the importance of assisting these schoolchildren educationally—considering each group or association’s location. It sought to find evidence of the importance of certain contextual elements in how students with rare diseases are schooled, since knowledge of this influence by all agents with an impact on that schooling may be a determining factor in learning processes, and more inclusive teaching and learning for these schoolchildren.

## 2. Materials and Methods

### 2.1. Participants

This study looked at a sample of teachers and a sample of rare disease support groups (associations). Participants were selected at random based on expressing an interest in participating through contact with the schools or groups.

A total of 574 teachers from schools in the autonomous community of Castilla-La Mancha participated. The distribution of the sample is as follows:School type: public (501 or 87.28%) or private–independent (73 or 12.72%). Public schools are fully funded by the regional education authority, but private–independent schools receive no public funding (private) or partial public funding (concertados) but are educationally independent.Location/setting: urban (347 or 60.45%) or rural (227 or 39.55%). Urban means towns or cities with more than 10,000 inhabitants, rural means towns with fewer than 10,000 inhabitants.

The percentages for teachers in relation to these two variables are from the Spanish Ministry of Education and Professional Training [35].

A total of 152 Spanish rare disease advocacy groups participated out of the 406 registered in the FEDER [29]. Most of the associations were regional or national in terms of their physical location or registered offices. Only 8 (5.26%) of the associations were in a rural setting, compared to 144 (94.74%) which were in urban settings.

### 2.2. Instruments

Two instruments were created ad hoc for the study in accordance with item response theory: a questionnaire aimed at teachers (see Table 1) and one aimed at advocacy groups for students with rare diseases. The teacher questionnaire evaluated their knowledge and thoughts about rare diseases. It was structured as in Table 1.

The patient group questionnaire assessed the actions they take to raise awareness and publicize their work in the educational field. This questionnaire is shown in Table 2.

Both questionnaires had similar numbers of items and dimensions, and a similar conceptual basis, although each was tailored to its target sample. The response options for both questionnaires ranged from 1 to 5: not at all (1), a little (2), moderately (3), quite a bit (4), and a lot (5). The final section of each questionnaire asked for observations and comments to collect qualitative information to round out the quantitative approach of the full research process. The results of the statistical validation of the two questionnaires are given below (Table 3). Data analysis was undertaken with the statistical program SPSS (including software Amos) and data interpretation was aided by a panel of 7 experts with a broad practical knowledge of educational inclusion.

### 2.3. Procedure and Data Analysis

The study was carried out during 2021 and 2022. To collect the information, the study material was sent to all schools in Castilla-La Mancha and to all rare disease advocacy groups listed in Spain [29], asking them to participate in the study. The information collected from the questionnaires was anonymous and confidential, and the participants had the option to be informed of the results. The data were analysed using SPSS version 28 and processed by calculating the descriptive statistics and comparisons of the means tests (Student’s *t*-test).

## 3. Results

This section is divided into three parts: the results from teachers according to the school type and location/setting, results from patient advocacy groups for rare diseases, and a comparison of those two sets of results.

### 3.1. Teachers’ Knowledge and Perceptions According to School Type and Location/Setting

Before looking at the results of teachers’ knowledge and perceptions according to the school type and location/setting, the mean results for each item and dimension in the instrument are given in Figure 1.

The highest mean scores were for item 20 (M = 4.72, SD = 0.71), item 13 (M = 4.50, SD = 0.89), and item 9 (M = 4.48, SD = 0.82). Participating teachers highlighted the importance of having a close relationship with their students’ families, whether in relation to diagnostic processes or educational intervention. In addition, they demonstrated a strong predisposition to seek information about the characteristics and needs of the students in their classrooms, something that most of them did of their own volition. They were also aware of the current legislation about providing an inclusive education to students with educational needs, and aware of the lack of specific regulatory frameworks for these children with rare diseases. In contrast, the lowest scoring items were item 18 (M = 1.97, SD = 1.15), item 2 (M = 2.03, SD = 1.11), and item 3 (M = 2.23, SD = 1.28), resulting from the number and variety of rare diseases, the small numbers present in schools, difficulties in generalizing specific interventions from one child to another and, in general, weak teacher training. These are fundamental aspects to be taken into consideration in teaching and learning processes. These results are in line with the conclusions from other studies on this subject [14,17].

In terms of dimensions, the mean scores were as follows: 13.85 (SD = 4.74) for conceptualization; 20.03 (SD = 4.10) for legislation; 16.27 (SD = 5.40) for intervention; and 16.33 (SD = 4.22) for diagnosis. The highest scoring dimension was legislation, which reflects the teachers’ knowledge of the essential regulatory frameworks that legally support the educational inclusion of all students with educational needs, although they indicated the importance of extending this regulation to action protocols and coordination for the education of students with rare diseases. In contrast, the lowest mean score was in the conceptualisation dimension, which reflects the need to increase efforts in teacher training in this area. This emphasises the importance of local and regional administrations in this regard and in the effective distribution of material and human resources to deal with students with rare diseases. There needs to be a multidimensional coordination between the internal and external services to schools, whether educational or otherwise, highlighting the multidimensional approach to the psychosocial, educational, and health elements of the affected individuals [36], and the key aspects in their adaptation, satisfaction, and experience of subjective well-being [37]. The overall mean for the instrument was 66.49 (SD = 16.41), with a negative asymmetry of −0.19 and a negative or platykurtic kurtosis of −0.82.

The results according to the school type are show below (see Table 4).

Table 4 shows that there were statistically significant differences, most at *p* < 0.001, in scores between the two types of school in thirteen of the twenty items, the four dimensions, and the overall score. The effect sizes were moderate to large [38]. In every case, teachers at publicly funded schools scored higher, something to be taken into consideration in the generalization of training activities aimed at teachers. It is worth noting that all the items in the legislation dimension exhibited statistically significant differences, although any planned training activities should cover all the dimensions.

The results based on the school location/setting are shown in Table 5.

Table 5 indicates that there were few statistically significant differences in scores according to school location/setting, specifically in item 2 with *p* = 0.037, item 6 with *p* = 0.020, item 11 with *p* = 0.001, and item 20 with *p* = 0.002, with moderate effect sizes [38]. In items 2 and 11, teachers at schools in urban settings scored higher, whereas in items 6 and 20, teachers in rural schools had higher scores. There was no pattern in the difference in the results between the teachers from one school setting to another. There were no statistically significant differences between the scores in the four dimensions or the overall score.

### 3.2. Results According to the Patient Advocacy Group Location/Setting

The results of the questionnaire for patient advocacy group are shown in Figure 2.

The highest mean scores were in items 11 (M = 4.39, SD = 0.75), 16 (M = 4.32, SD = 0.97), and 20 (M = 4.30, SD = 0.95). The associations do publicize their rare diseases in schools, emphasizing the characteristics and warning signs, and demonstrate to encourage research in order to achieve more effective and better targeted (as well as more widely available and cheaper) pharmacological treatments and interventions [39,40]. In contrast, the lowest scores were in item 3 (M = 3.02, SD = 1.42), item 17 (M = 3.13, SD = 1.29), and item 2 (M = 3.21, SD = 1.41), resulting from not including the prevalence of existing cases and types of rare diseases in training in schools, and not presenting examples of specific cases already detected with a tailored educational response.

The mean scores for the dimensions were: 16.32 (SD = 5.75) for the conceptualization; 19.20 (SD = 4.13) for legislation; 20.99 (SD = 3.39) for intervention; and 16.43 (SD = 3.74) for diagnosis. The highest scoring dimension was intervention, which is logical given that healthcare is a fundamental part of the associations’ day-to-day activities. The lowest mean score was for the conceptualisation dimension, which indicates a need for increased efforts in teacher training with dissemination and outreach tasks that include the characteristics, typology, etiology, and prevalence of rare diseases, fundamental to promoting these children’s educational and social inclusion. The overall mean score for the instrument was 72.52 (SD = 13.01), with a negative asymmetry of −0.56 and a positive or leptokurtic kurtosis of 0.39.

The results according to the location/setting of the association are shown in Table 6.

These results need to be taken with caution because of the very different sample sizes between the two settings. The vast majority (144; 94.74%) of patient advocacy groups were based in urban settings, compared to eight (5.26%) in rural settings. Associations need to address rural locations because once families have members with a rare disease, they seek places where they can raise their concerns and questions, a need that is not always met by the health authorities. Establishing links to other families in other places is vital. The rural-based associations that participated stressed the need to strengthen their networks and ties with other social groups in order to be more effective.

As Table 6 indicates, there were only statistically significant differences in one item, item 3, with *p* = 0.023, and a large effect size of 0.83 [38]. Rural-based associations had a higher mean score, 4.13 (SD = 0.83), than urban based associations, 2.96 (SD = 1.42). In the dimensions, there were higher mean scores for rural-based associations in conceptualisation and legislation, and for urban locations in intervention and diagnosis.

In short, it is essential for associations to broaden their educational and social contexts, transmitting the distinctive characteristics of the diseases they deal with is key to raising the profile of the work they do [21]. One valuable way to raise awareness of what their members need, their characteristics, and their care, may be demonstrating their realities through the stories of affected people and their families. The associations indicated that they do this in various ways, such as giving talks to teachers, through the media, solidarity sporting events, information campaigns taking advantage of annual events, and attendance at specific training sessions, if they participate in the Training Cycle of a Higher Degree of Social Integration in secondary education schools. This approach could have a positive impact on what associations consider to be key aspects, such as the visibility and promotion of research to achieve advances in this field of rare diseases.

### 3.3. Comparison of Results between the Two Questionnaires

Although the two questionnaires had similar numbers of items and dimensions, and share the objective of assessing knowledge and perceptions, there were some differences in terms of the target audiences. Despite that, we think it is worth comparing the results. In both questionnaires, item 20 was scored very highly, so both groups were sensitive to the importance of dedicating more resources to research in the field of rare diseases. In addition, items 2 and 3 in both questionnaires were among the lowest mean scores, so the typology and prevalence of rare diseases should be included in training sessions in schools.

The dimensions produced the following mean scores in the two questionnaires (see Figure 3).

Figure 3 suggests two patterns. Associations had higher mean scores than teachers in the conceptualisation and intervention dimensions (+2.47 and +4.72 points, respectively). The means for the legislation and diagnosis dimensions were very similar between the two groups. Finally, looking at the overall scores for the questionnaires, the associations had a higher mean score of 72.52 (SD = 13.01) than the teachers at 66.49 (SD = 16.41). Both questionnaires had a negative asymmetry, whereas kurtosis was negative in the teachers’ questionnaire and positive in the associations’ questionnaire.

## 4. Discussion

People with rare diseases have reported difficulties in getting a diagnosis, implementing treatment, or taking advantage of advances in research that could affect them [39,40,41]. This seriously affects their quality of life and their personal development, so urgent measures are needed to address these deficiencies. To advance towards these objectives, measures should be taken in all societal contexts. In education, there should be specific training activities for teachers about rare diseases, class sizes in schools should be reduced, protocols and action guides should be developed, and there should be the increased provision of personal and material resources, among other things. In the social sphere, it would be useful to support the care and lobbying sides of patient advocacy groups, and to cover areas with lower population densities. Finally, in healthcare, waiting times could be reduced, healthcare personnel increased, and improved communication channels could be established with schools for a better coordination and to support an early differential diagnosis [19].

This study examined the contextual factors that could have an impact on the above from two perspectives: teachers and patient advocacy groups. The results from the teachers’ questionnaire reflects the importance of a close communication with these students’ families in all aspects of the teaching and learning process. In addition, teachers want to help these children educationally, modifying their teaching practices according to the current legal frameworks (the legislation dimension had the highest average scores, with a mean of 20.03). In contrast, they indicated the need to expand their training in this area, demonstrating ignorance in several elements characterizing the conceptualisation of rare diseases and noting difficulties in generalizing specific actions from one case to another due to the number and variety of rare diseases (the conceptualisation dimension had the lowest average results, with a mean of 13.85).

Teachers working at publicly funded schools had statistically significantly higher scores in most items, in all of the dimensions, and overall, the most at *p* < 0.001. In contrast, there were only statistically significant differences between urban and rural schools in four items (2, 6, 11, and 20, one item from each of the four dimensions), but there was no clear pattern of higher scores for one or the other.

Patient advocacy groups had higher scores in their questionnaires in the intervention and legislation dimensions (mean values of 20.99 and 19.20, respectively), and lower scores in the conceptualisation and diagnosis dimensions (mean values of 16.32 and 16.33, respectively). Although they already do training activities in schools, associations should focus on that, including content related to the types and prevalence of rare diseases or knowledge of their warning signs, preferably with specific case examples. There was a statistically significant difference between rural- and urban-based associations in only one item—in the conceptualisation dimension (item 3)—with rural-based associations scoring higher (value of 4.13), with a large effect size (0.83). Rural-based associations scored higher in the conceptualisation and legislation dimensions (mean scores of 17.75 and 19.63, respectively) and urban-based associations scored higher in the intervention and diagnosis dimensions (mean scores of 21.03 and 19.48, respectively).

When comparing the results from the two questionnaires, it is worth highlighting item 20, emphasizing that both the teachers and associations consider research essential to improve these children’s well-being. Both groups indicated the need to include knowledge of the typology, prevalence, and warning signs of these rare diseases in the training processes. The results of this study show the importance of associations providing training to improve conceptualisation and educational intervention for these children, which needs schools to be open to this training.

The participation of teachers in these training activities will be essential in addressing the concerns, knowledge, and perceptions communicated by members of these associations. Raising awareness of rare diseases is essential [16] for reducing the inequalities suffered by this group [42]. One example is the ‘home delivery’ service provided by one of the participating associations (Federación Española de Hemofilia, FEDHEMO). This service facilitated by the Government of the Community of Madrid delivers the medication these people need directly from hospitals to their homes to minimise the disruption to their daily work or school routines.

In line with this, it is important to remember that each student is unique and educational systems should promote educational practices that address their characteristics, potentials and needs. Although students with rare diseases are a minority in the classroom and the breadth of diseases means a wide variety of educational interventions [14], the participating associations are aware of this situation and have stated that this should not lead to discrimination and segregation. In this regard, a more widespread dissemination of manuals and action guides that show the characteristics, typology, aetiology, prevalence and actions, and successful intervention proposals, among other things, may be an interesting starting point for improving educational inclusion for these schoolchildren with rare diseases [43].

## 5. Conclusions

It is essential to seek meeting points to exchange information and experiences between all contexts with an impact on the schooling of students with rare diseases. These links promote mutual trust and progress in accordance with the established goals, since the experience of the disease in the personal and family spheres has notable implications on educational, social, and health contexts [11]. This constitutes one of the principal engines for coping with these diseases and promoting educational and social changes in the reality of these students.

One of the main limitations of the study is the discrepancy between the sample sizes of teachers from publicly funded and private–independent schools, although the difference is in line with the current data from the community of Castilla-La Mancha. A similar limitation applies to the selection of urban- and rural-based associations. Improving the balance of the sample in terms of these variables will be a priority objective when replicating these results in future research. In addition, future studies will include a regression and correlation analysis to complete the statistical study.

In summary, there is a pressing need to promote the dissemination of knowledge about these rare diseases, and greater community and institutional involvement is essential. The fact that they are incurable and chronic, together with an ignorance about many of them, underscores the need to dedicate greater resources to research into and funding for these rare disease patient advocacy groups [44]. Research such as the present study should make it possible to raise the profile of the characteristics and needs of this group and their families. The ultimate goal is to improve their quality of life and well-being, and allow educational, social, and cultural inclusion.

## Figures and Tables

**Figure 1 ijerph-19-14103-f001:**
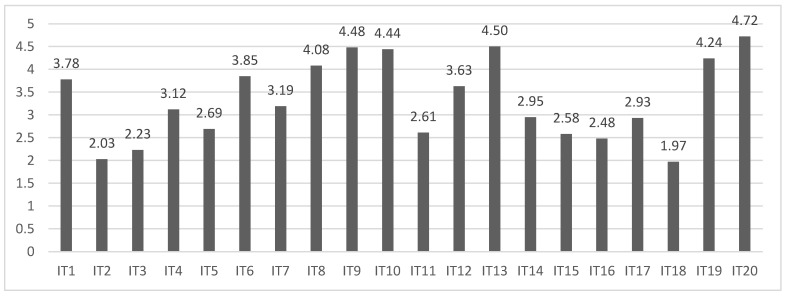
Mean scores for each item in the teachers’ questionnaire.

**Figure 2 ijerph-19-14103-f002:**
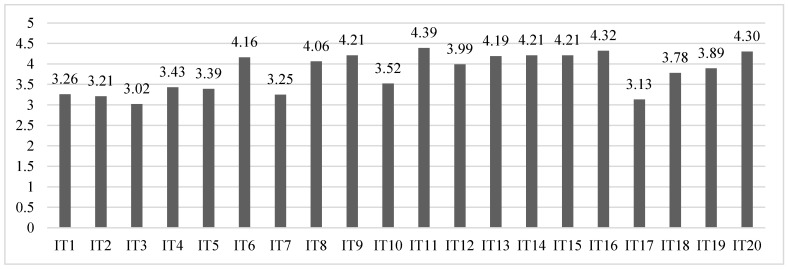
Mean scores for each item in the patient advocacy group questionnaire.

**Figure 3 ijerph-19-14103-f003:**
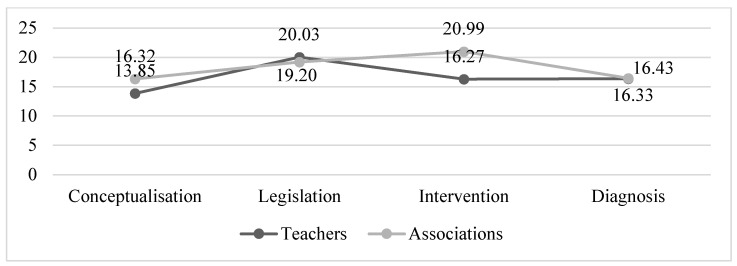
Mean scores for each dimension in the teachers and patient advocacy group questionnaires.

**Table 1 ijerph-19-14103-t001:** Structure of the teachers’ questionnaire.

Dimension	Items
Conceptualisation	1. I know what a rare disease is.
	2. I know the categories that the different types of rare diseases fall into.
	3. I know how prevalent the rare diseases I have dealt with in my school are at a national or international level.
	4. When I have had a student with a rare disease in my class, I knew what their main characteristics were.
	5. I have maintained contact from the school with patient advocacy groups that deal with rare diseases.
Legislation	6. I know the educational inclusion legislation in our region.
	7. I know the content of the most recent education legislation, especially about student educational needs
	8. I have read and understand the article in the Spanish Constitution which describes the right to education for all.
	9. I understand the importance of there being a law aimed at educational inclusion
	10. Within my teaching, I believe inclusive education to be important, and therefore the principles of inclusion must be applied in the classroom.
Intervention	11. I have had, or still have, a student in my class with a rare disease.
	12. I have educated myself about the signs and symptoms students may exhibit if they suffer from a rare disease.
	13. I think the family–school relationship is essential for proper intervention for students with rare diseases.
	14. I know specific activities to do with these students.
	15. I am able to advise and guide other teachers about activities with these students.
Diagnosis	16. I know the warning signs a student with a possible rare disease may present.
	17. I educate myself and try to contact other diagnosed cases in order to improve the education I give to students with rare diseases that I might encounter as a teacher.
	18. I know how to clinically diagnose a rare disease.
	19. I think the relationship between the school and the families is essential in order to be able to properly detect and help an initial diagnosis.
	20. I am aware of the need for more research to help diagnosis and treatment.

**Table 2 ijerph-19-14103-t002:** Structure of the patient advocacy group questionnaire.

Dimension	Items
Conceptualisation	1. We provide information about the conceptualisation of the rare diseases we represent.
	2. We indicate the category our disease falls within.
	3. Schools are shown the prevalence of our disease.
	4. The distinctive characteristics of our disease are explained.
	5. From the association, we coordinate with schools in relation to our disease.
Legislation	6. On behalf of the association, we are aware of the Decrees of Educational Inclusion and/or Attention to Diversity in our region.
	7. We know the content of the latest Organic Law on Education on everything related to the educational needs of students.
Legislation	8. We have read and are aware of the article in the Spanish Constitution indicating the right to education for all.
	9. The association publicises the importance of there being a Decree for Educational Inclusion/Attention to Diversity.
	10. The association insists on the need to apply the principles of Educational Inclusion/Attention to Diversity outlined the legislation to the people in our group.
Intervention	11. The work with rare diseases is shown from the associative field.
	12. Our association investigates the symptoms and characteristics in order to intervene more effectively.
	13. We work with families and schools to offer educational guidelines on what interventions they should perform for children.
	14. Specific actions that can be carried out with this group in the classroom are publicised.
	15. Guidance and advice is offered to these children’s teachers.
Diagnosis	16. Schools are informed about the warning signs that a person affected by a rare disease could have.
	17. From the association, individual diagnoses of each of the rare diseases that we treat are disclosed.
	18. We show schools the protocols to follow to make a clinical diagnosis.
	19. We highlight the importance of the family–school relationship for adequate initial diagnosis.
	20. We emphasise the need for research so that the diagnosis is increasingly accurate and appropriate and less expensive for each person.

**Table 3 ijerph-19-14103-t003:** Statistical validation of the two questionnaires.

Statistics	Indexes	
	Teachers	Associations
Cronbach’s Alpha	Between 0.79 and 0.94	Between 0.77 and 0.88
Content validity	Between 0.77 and 0.94	Between 0.79 and 0.86
Kappa Index	0.88	0.86
Exploratory factor analysis (EFA):		
KMO	0.94	0.82
Bartlett’s sphericity test	7699.08	1838.05
Significance	*p* < 0.001	*p* < 0.001
Percentage of total variance explained	68.49%	65.04%
Factorial structure	4 factors	4 factors
Confirmatory factor analysis (CFA):		
CMIN	32.48	38.23
*p*	0.09	0.08
RMSEA	0.05	0.04
CFI	0.95	0.93
TLI	0.94	0.92
Factorial structure	4 factors	4 factors

**Table 4 ijerph-19-14103-t004:** *T*-test based on the variable school type.

IT/D	Ownership				t	df	*p*	d
	Public		Private–Independent					
	M	SD	M	SD				
IT1	3.86	1.07	3.23	1.21	4.56	572	<0.001 ***	0.57
IT2	2.05	1.12	1.88	1.03	1.27	572	0.203	0.16
IT3	2.28	1.29	1.92	1.13	2.24	572	0.025 *	0.28
IT4	3.15	1.26	2.90	1.17	1.57	572	0.117	0.20
IT5	2.70	1.34	2.63	1.38	0.43	572	0.667	0.05
D1	14.04	4.69	12.56	4.93	2.50	572	0.013 *	0.31
IT6	3.90	1.17	3.56	1.24	2.26	572	0.024 *	0.28
IT7	3.27	1.44	2.60	1.57	3.67	572	<0.001 ***	0.46
IT8	4.12	1.08	3.79	1.10	2.39	572	0.017 *	0.30
IT9	4.55	0.74	3.99	1.10	5.64	572	<0.001 ***	0.71
IT10	4.49	0.81	4.05	1.09	4.13	572	<0.001 ***	0.52
D2	20.33	3.93	18.00	4.66	4.62	572	<0.001 ***	0.58
IT11	2.59	1.53	2.75	1.41	0.84	572	0.398	0.11
IT12	3.71	1.44	3.07	1.53	3.53	572	<0.001 ***	0.44
IT13	4.58	0.79	3.93	1.28	6.01	572	<0.001 ***	0.75
IT14	2.99	1.33	2.67	1.41	1.91	572	0.056	0.24
IT15	2.63	1.42	2.26	1.48	2.05	572	0.041 *	0.26
D3	16.50	5.22	14.68	6.31	2.70	572	0.007 **	0.34
IT16	2.52	1.22	2.22	1.23	1.97	572	0.050	0.25
IT17	2.98	1.31	2.56	1.34	2.53	572	0.012 *	0.32
IT18	1.98	1.16	1.85	1.08	0.93	572	0.350	0.12
IT19	4.35	1.03	3.48	1.50	6.28	572	<0.001 ***	0.79
IT20	4.81	0.56	4.11	1.20	8.36	572	<0.001 ***	1.05
D4	16.64	3.93	14.22	5.40	4.67	572	<0.001 ***	0.58
Total	67.52	15.63	59.47	19.70	3.97	572	<0.001 ***	0.50

* Significant at 5% (*p* < 0.05). ** Significant at 1% (*p* < 0.01). *** Significant at 0.01% (*p* < 0.001).

**Table 5 ijerph-19-14103-t005:** *T*-test based on the variable school location/setting.

IT/D	Location/Setting				t	df	*p*	d
	Urban		Rural					
	M	SD	M	SD				
IT1	3.75	1.14	3.81	1.06	−0.66	572	0.508	0.06
IT2	2.11	1.17	1.91	1.01	2.09	572	0.037 *	0.18
IT3	2.23	1.28	2.22	1.27	0.08	572	0.936	0.01
IT4	3.14	1.23	3.08	1.27	0.54	572	0.590	0.05
IT5	2.72	1.33	2.66	1.36	0.53	572	0.593	0.05
D1	13.95	4.88	13.69	4.53	0.65	572	0.517	0.05
IT6	3.76	1.24	4.00	1.08	−2.33	572	0.020 *	0.20
IT7	3.14	1.49	3.26	1.45	−0.88	572	0.377	0.08
IT8	4.08	1.12	4.08	1.05	−0.02	572	0.987	0.00
IT9	4.42	0.88	4.56	0.71	−1.95	572	0.052	0.17
IT10	4.40	0.89	4.49	0.81	−1.27	572	0.206	0.11
D2	19.81	4.22	20.38	3.89	−1.65	572	0.099	0.14
IT11	2.79	1.52	2.34	1.47	3.54	572	<0.001 ***	0.30
IT12	3.66	1.43	3.58	1.52	0.63	572	0.531	0.05
IT13	4.48	0.93	4.53	0.84	−0.72	572	0.473	0.06
IT14	2.95	1.35	2.96	1.32	−0.07	572	0.946	0.01
IT15	2.59	1.45	2.56	1.41	0.28	572	0.780	0.02
D3	16.47	5.52	15.97	5.22	1.09	572	0.275	0.09
IT16	2.50	1.24	2.45	1.20	0.52	572	0.600	0.04
IT17	2.91	1.30	2.94	1.34	−0.26	572	0.796	0.02
IT18	2.04	1.17	1.86	1.12	1.82	572	0.069	0.15
IT19	4.19	1.17	4.30	1.10	−1.14	572	0.255	0.10
IT20	4.65	0.79	4.84	0.54	−3.13	572	0.002 **	0.27
D4	16.30	4.36	16.39	4.00	−0.26	572	0.792	0.02
Total	66.53	17.07	66.44	15.37	0.07	572	0.946	0.01

* Significant at 5% (*p* < 0.05). ** Significant at 1% (*p* < 0.01). *** Significant at 0.01% (*p* < 0.001).

**Table 6 ijerph-19-14103-t006:** *T*-test based on the variable association location/setting.

IT/D	Setting				t	df	*p*	d
	Urban		Rural					
	M	SD	M	SD				
IT1	3.26	1.50	3.38	0.74	−0.22	150	0.826	0.08
IT2	3.19	1.43	3.50	0.93	−0.59	150	0.553	0.22
IT3	2.96	1.42	4.13	0.83	−2.29	150	0.023 *	0.83
IT4	3.40	1.43	4.00	1.07	−1.17	150	0.242	0.43
IT5	3.43	1.17	2.75	0.89	1.62	150	0.107	0.59
D1	16.24	5.86	17.75	3.01	−0.72	150	0.470	0.26
IT6	4.14	0.93	4.63	0.52	−1.46	150	0.145	0.53
IT7	3.25	1.21	3.25	1.39	0.00	150	1.000	0.00
IT8	4.07	1.09	3.88	0.99	0.49	150	0.622	0.18
IT9	4.19	0.85	4.63	0.52	−1.43	150	0.154	0.52
IT10	3.53	1.35	3.25	1.16	0.58	150	0.561	0.21
D2	19.18	4.20	19.63	2.77	−0.29	150	0.768	0.11
IT11	4.37	0.75	4.75	0.46	−1.41	150	0.160	0.51
IT12	4.01	1.10	3.75	1.28	0.64	150	0.525	0.23
IT13	4.21	0.91	3.88	0.83	1.01	150	0.316	0.37
IT14	4.23	0.94	3.88	0.83	1.04	150	0.301	0.38
IT15	4.22	1.02	4.00	1.07	0.60	150	0.551	0.22
D3	21.03	3.46	20.25	1.67	0.63	150	0.526	0.23
IT16	4.33	0.95	4.13	1.36	0.59	150	0.558	0.21
IT17	3.17	1.31	2.50	0.53	1.43	150	0.156	0.52
IT18	3.80	1.10	3.50	1.07	0.75	150	0.456	0.27
IT19	3.87	1.08	4.38	1.19	−1.28	150	0.203	0.46
IT20	4.31	0.93	4.00	1.31	0.91	150	0.366	0.33
D4	19.48	3.81	18.50	1.77	0.72	150	0.473	0.26
Total	72.60	13.30	71.00	5.85	0.34	150	0.736	0.12

* Significant at 5% (*p* < 0.05).

## Data Availability

Due to the anonymity and confidentiality of the data obtained, the authors have not reported any of the data obtained, the purpose of which is exclusively the development of this research.

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
