# Peer review of "Contextual Variables with an Impact on the Educational Inclusion of Students with Rare Diseases"

_ijerph, 2022, doi:10.3390/ijerph192114103_

Round 1
Reviewer 1 Report
1. The hypotheses of the study should be stated at the end of the introduction.
2. If, as it appears, the questionnaires used in the study were ad hoc, this should be indicated.
3. The information on data analysis is incomplete. Table 3 presents analysis results that are not indicated. Has a CFA been performed? In what way? With what software? Has an EFA been previously performed? Has a previous study been conducted to validate the questionnaires? Is there previous literature to support construct validation? How have the dimensions of the questionnaires been established?
4. It is said that "comparisons of mean tests" are performed. What kind of tests have been performed? Parametric testing? Non-parametric? Which ones? Have the assumptions been checked? It is understood by the results that a Student's T test has been performed.
5. Table 4 lacks information on statistical significance at the bottom.
6. The type of experimental design should be indicated as a limitation, as well as the use of unvalidated instruments (if applicable) and the type of analyses carried out. A regression study, for example, has not been performed, nor has the effect size or required sample size been calculated. In the design and validation of questionnaires (if applicable), it should be indicated whether the Item Response Theory or the Classical Test Theory has been followed. Similarly, the required number of subjects should be analyzed in terms of the total number of items and correlation analysis should be performed.
Author Response
Dear Reviewer,
We appreciate the review work done, thank you very much for considering our work. The following is a response to each of the reviews carried out.
Best regards
---
Comments and Suggestions for Authors
Reviewer. 1. The hypotheses of the study should be stated at the end of the introduction.
Authors. It includes: “It sought to find evidence of the importance of certain contextual elements in how stu-dents with rare diseases are schooled, since knowledge of this influence by all agents with an impact on that schooling may be a determining factor in learning processes, and more inclusive teaching and learning for these schoolchildren”.
Reviewer. 2. If, as it appears, the questionnaires used in the study were ad hoc, this should be indicated.
Authors. Review is included.
Reviewer. 3. The information on data analysis is incomplete. Table 3 presents analysis results that are not indicated. Has a CFA been performed? In what way? With what software? Has an EFA been previously performed? Has a previous study been conducted to validate the questionnaires? Is there previous literature to support construct validation? How have the dimensions of the questionnaires been established?
Authors. An EFA and a CFA have been carried out. The first has been carried out using SPSS version 28 and the second using the AMOS program. As a preliminary study, a panel of 7 experts with broad practical knowledge of educational inclusion, four of whom had specific training in rare diseases, was carried out, serving to calculate the content validity index and establish, in advance, the dimensions of the instruments.
Reviewer. 4. It is said that "comparisons of mean tests" are performed. What kind of tests have been performed? Parametric testing? Non-parametric? Which ones? Have the assumptions been checked? It is understood by the results that a Student's T test has been performed.
Authors. It is included that comparisons of means have been made using Student's t-test.
Reviewer. 5. Table 4 lacks information on statistical significance at the bottom.
Authors. Information on significance is included.
Reviewer. 6. The type of experimental design should be indicated as a limitation, as well as the use of unvalidated instruments (if applicable) and the type of analyses carried out. A regression study, for example, has not been performed, nor has the effect size or required sample size been calculated. In the design and validation of questionnaires (if applicable), it should be indicated whether the Item Response Theory or the Classical Test Theory has been followed. Similarly, the required number of subjects should be analyzed in terms of the total number of items and correlation analysis should be performed.
Authors. It is included as a proposal to incorporate in future research the realization of regression and correlation analysis. The size of the effect has been calculated (d). It is included in the text "in accordance with Item Response Theory".

Reviewer 2 Report
I read the article with great interest. It is an extension of the authors' interest in the inclusive education for students with rare diseases, in a given region in Spain. By comparison to prior articles published by this research team, new angles and aspects of the topic are presented and extend the knowledge on the challenges inclusive education pose in the European educational practice. While the article is interesting in topic and has some valuable results, it is difficult to read due to a low level of English. A professional proofreading and correction is crucial to ensure the clarity and readability of the article. Besides linguistic aspects, authors should pay attention and improve some aspects in the presentation of their research.
1. In the Abstract, the last sentence is non-informative. The authors state that "In short, numerous variables that influence teaching and learning processes need to be considered in educational praxis, and this is essential for increasingly heterogeneous educational locations that demand multidimensional approaches." It is too broad and vague. What approaches? What are those variables and why do they have to be considered?
2. In the Introduction part, page 2 (bottom) authors refer to the United Nations and then jump to the Spanish situation too abruptly. How about a European perspective? I would suggest, for instance, (2019) Schooling of Children with Rare Diseases and Disability in Europe, International Journal of Disability, Development and Education, 66:4, 362-373, DOI: 10.1080/1034912X.2018.1562159. With the analysis of the situation in 8 countries it would help getting a better picture on how the Spanish situation resonates/differs from other European countries.
3. In the section Materials and methods, subsection Instruments, Table 3 and the comments belong to the Results, not to the instruments. I understand that the authors want to explain the validity of the selected instruments, but there is not enough clarity around this topic. Also, more information about the questionnaires is necessary before introducing them to the reader: how they were designed, are they similar to other questionnaires used for investigating this topic etc.
4. The Discussion section needs to be extended, to explain and debate the results presented synthetically in a numerical form in the previous section. Sentences like “Education professionals must consider the concerns, knowledge and perceptions raised by members of these associations” are not necessarily supported by the results. Further the given example with the ‘home delivery’ service provided by one of the participating associations should be better introduced and explained. Is it a singular recommendation? Are there other proposals worthy of considering in organizing the schooling of children with special needs?
5. The Conclusions section also contains Limitations. It would be better to split the idea, and not mix conclusions with weaknesses and further study intentions.
I believe that the article has a good potential and that a better presentation could help promote the valuable results obtained during the research.
Author Response
Dear Reviewer,
We appreciate the review work done, thank you very much for considering our work. The following is a response to each of the reviews carried out.
Best regards
---
Comments and Suggestions for Authors
Reviewer. It is difficult to read due to a low level of English. A professional proofreading and correction is crucial to ensure the clarity and readability of the article. Besides linguistic aspects, authors should pay attention and improve some aspects in the presentation of their research.
Authors. The article has been revised by a native translator. Also has been reviewed by a native translator, we are very sorry for the inconvenience caused.
Reviewer. 1. In the Abstract, the last sentence is non-informative. The authors state that "In short, numerous variables that influence teaching and learning processes need to be considered in educational praxis, and this is essential for increasingly heterogeneous educational locations that demand multidimensional approaches." It is too broad and vague. What approaches? What are those variables and why do they have to be considered?
Authors. In the abstract, it is included "in this study we looked at those of a contextual nature".
Reviewer. 2. In the Introduction part, page 2 (bottom) authors refer to the United Nations and then jump to the Spanish situation too abruptly. How about a European perspective? I would suggest, for instance, Renata Linertová, Javier González-Guadarrama, Pedro Serrano-Aguilar, Manuel Posada-De-la-Paz, Márta Péntek, Georgi Iskrov & Marta Ballester (2019) Schooling of Children with Rare Diseases and Disability in Europe, International Journal of Disability, Development and Education, 66:4, 362-373, DOI: 10.1080/1034912X.2018.1562159. With the analysis of the situation in 8 countries it would help getting a better picture on how the Spanish situation resonates/differs from other European countries.
Authors. Included. “At European level, it is worth highlighting the European Solidarity Corps [31], since its priorities include social inclusion, valuing difference and diversity for all, and equal ac-cess to opportunities through necessary resources, activities and support measures. Ac-cepting these goals at the European level is essential to join actions in the implementation of inclusive education in Europe [32]”.
Reviewer. 3. In the section Materials and methods, subsection Instruments, Table 3 and the comments belong to the Results, not to the instruments. I understand that the authors want to explain the validity of the selected instruments, but there is not enough clarity around this topic. Also, more information about the questionnaires is necessary before introducing them to the reader: how they were designed, are they similar to other questionnaires used for investigating this topic etc.
Authors. An EFA and a CFA have been carried out. The first has been carried out using SPSS version 28 and the second using the AMOS program. As a preliminary study, a panel of 7 experts with broad practical knowledge of educational inclusion, four of whom had specific training in rare diseases, was carried out, serving to calculate the content validity index and establish, in advance, the dimensions of the instruments. In addition, it includes “ad hoc for the study in accordance with Item Response Theory”.
Reviewer. 4. The Discussion section needs to be extended, to explain and debate the results presented synthetically in a numerical form in the previous section. Sentences like “Education professionals must consider the concerns, knowledge and perceptions raised by members of these associations” are not necessarily supported by the results. Further the given example with the ‘home delivery’ service provided by one of the participating associations should be better introduced and explained. Is it a singular recommendation? Are there other proposals worthy of considering in organizing the schooling of children with special needs?
Authors. The Discussion section is improved. Text is modified, it is included: "The participation of teachers in these training activities will be essential in addressing the concerns, knowledge and perceptions communicated by members of these associations. Raising awareness of rare diseases is essential [16] for reducing inequalities suffered by this group [42]. One example is the ‘home delivery’, service provided by one of the participating associations (Federación Española de Hemofilia -FEDHEMO-). This service facilitated by the Government of the Community of Madrid delivers medication these people need directly from hospitals to their homes to minimise disruption to daily work or school routines". The exposed service example is specified and improved.
Reviewer. 5. The Conclusions section also contains Limitations. It would be better to split the idea, and not mix conclusions with weaknesses and further study intentions.
Authors. Ideas are divided to improve the expression of the content.

Round 2
Reviewer 1 Report
Appropriate clarifications were provided to the questions asked in the previous phase.
Author Response
Dear Reviewer,
We greatly appreciate the review process developed, thank you very much for all your contributions.
Best regards,
Ramón

Reviewer 2 Report
Dear authors,
The article is interesting and gained clarity. Minor corrections would be necessary. For instance at pg. 9 after the table you inserted reference to (Pélissier et al., 2022; Reinhard et al., 2021). Most probably these would be references [38, 39]. Check again the numbering of references in the text and in the Reference list. At pg. 12, last paragraph "In line with this, it is important to remember that each student is unique and educational systems should promote educational practices that address their characteristics, potentials and eeds". The last word must be "needs".
Author Response

(The authors gave the same response as above.)
